# Modeling and Prediction of Water-Jet-Guided Laser Cutting Depth for Inconel 718 Material Using Response Surface Methodology

**DOI:** 10.3390/mi14020234

**Published:** 2023-01-17

**Authors:** Chuang Zhao, Yugang Zhao, Dandan Zhao, Qian Liu, Jianbing Meng, Chen Cao, Zhilong Zheng, Zhihao Li, Hanlin Yu

**Affiliations:** School of Mechanical Engineering, Shandong University of Technology, Zibo 255049, China

**Keywords:** water-jet-guided laser, surface quality, orthogonal experiments, RSM, Inconel 718, cutting depth

## Abstract

In this study, the water-jet-guided laser (WJGL) method was used to cut Inconel 718 alloy with high temperature resistance. The effect of critical parameters of the water-jet-guided laser machining method on the cutting depth was studied by a Taguchi orthogonal experiment. Furthermore, the mathematical prediction model of cutting depth was established by the response surface method (RSM). The validation experiments showed that the mathematical model had a high predictive ability for cutting depth. The optimal cutting depth was obtained by model prediction, and the error was 5.5% compared with the experimental results. Compared with the traditional dry laser cutting, the water conducting laser method reduced the thermal damage and improved the cutting quality. This study provides a reference for the precision machining of Inconel 718 with a water-jet-guided laser.

## 1. Introduction

Inconel 718 has excellent heat resistance, corrosion resistance, and high strength, as well as fatigue resistance and oxidation resistance. For example, in the operation of aircraft engines, turbine blades made of nickel-based superalloy materials enable the engine to maintain normal operating performance at temperatures as high as 1400 °C [1]. With the development of the aerospace and petrochemical fields, the application of nickel-based superalloys has become increasingly more extensive [2,3]. The weight of nickel-based superalloys accounts for more than 70% of aerospace engines. Therefore, the efficient processing and utilization of nickel-based superalloys has been considered by scientists. Inconel 718 has high shear strength, low thermal conductivity, and high strength, leading to its poor machinability. Traditional processing methods, such as turning, drilling, and milling, can cause serious wear of the tool. Moreover, in a high-temperature environment, the material has a chemical reaction tendency caused by the cutting force, and the surface finish after processing is poor [4]. At present, some traditional non-contact processing methods are used to process Inconel 718, such as electric discharge machining (EDM), electrochemical machining (ECM), and laser beam machining (LBM), which can bring high machining efficiency at a reasonable cost [5,6,7].

With the rapid development of the mechanical field, the requirements for the accuracy of material processing have also gradually increased, and the defects brought about by traditional processing methods have become increasingly apparent. For example, in aircraft engine turbine blade film cooling hole processing, there are extremely strict requirements for the surface quality of the film cooling hole. The traditional thermal laser processing will produce adverse side effects such as a recast layer, burr, micro-crack, slag, and heat-affected zone, and these minor side effects will seriously affect the engine performance and life [8]. This makes the machining of high-quality gas film holes extremely difficult in terms of achieving aerospace-grade high-precision requirements. Therefore, increasingly more researchers are investigating various novel material removal methods to achieve high-precision machining of high-temperature alloys [9,10,11].

In 1993, scientists in Switzerland demonstrated the feasibility of water as a medium to guide a laser for processing [12]. This new method couples a laser energy beam and a water jet to process the material, which can complete the removal of the material through the laser beam energy but also through the high-speed water jet to clean the material slag in a timely manner and complete the cooling of the cut, overcoming many of the undesirable side effects caused by conventional thermal laser processing. This laser cutting method with “fast cooling” and “automatic scouring” functions was first applied to the processing of semiconductor materials because of its advantages such as a small heat-affected zone and clean surfaces. Perrottet used the WJGL to process GaAs semiconductors, and the cooling and scouring of the water jet eliminated the heat-affected defects and achieved a clean, non-toxic treatment of the waste slag [13]. Moreover, in the medical field, WJGL has shown its great advantages. Wagner used the WJGL to cut vascular stent material with almost no heat-affected zone [14]. WJGL processing technology has become an effective tool for processing high-precision devices in microelectronics and medicine [15,16,17]. In recent years, WJGL processing technology has gradually started to be applied in the field of metal materials, especially in high-temperature-resistant aerospace alloys, where water-jet guided laser cutting technology has its unique processing advantages and application prospects [18,19,20]. Traditional laser cutting will cause thermal damage to the metal surface and produce a white layer. When a laser beam is used to machine nickel-base superalloy, the mechanism of the white layer is the crystal recasting caused by the thermomechanical effect of the laser at high temperature. In water-jet-guided laser processing, the absorption of heat inside the material by water jet can reduce the thermomechanical effect, thus greatly reducing the occurrence of white layer and crystal recasting and other undesirable phenomena.

In order to solve the problem of large thermal damage of conventional laser cutting Inconel 718, we built a water-guided laser processing platform and used the water-jet-guided laser method to perform cutting experiments on Inconel 718 material. In this paper, firstly, the effects of water pressure, laser power, laser frequency, and feed speed on the cutting depth were studied by orthogonal experiments. Finally, the building blocks of the cutting depth prediction model were completed by response surface methodology (RSM), and the RSM was used again to find the optimal combination of process parameters. This study achieved a controlled process for water-jet-guided laser cutting Inconel 718 processing, which is a guideline for future research.

## 2. Experimental Equipment

The water-guided laser processing equipment in the experiment is shown in Figure 1. It is mainly composed of a laser generator, a water-laser coupling system, a computer control system, a water circulation system, and a processing platform. The quasi-continuous fiber laser (YLR-2000-WC) has a laser wavelength of 1070 nm, maximum power of 2000 W, and maximum pulse repetition frequency of 10,000 Hz. The diameter of the water jet nozzle used in this experiment was 500 μm, because the nozzle exit was at an 90° angle, and there will be shrinkage after the high-speed water jet is emitted; the actual diameter of the water jet was about 400~450 μm, and the diameter of the water jet determined the diameter of the laser energy beam. To ensure the quality of water-light coupling, filtered deionized water was used for the experiments. The water-jet-guided laser processing platform is shown in Figure 2.

The experimental material was an Inconel 718 plate, as shown in Figure 3a. The size was 50 × 50 × 1 mm; it was provided by Dongguan Tongheng Materials Co., Ltd., Dongguan, China; and its chemical composition is shown in Table 1. The water-jet-guided laser equipment was used for linear scanning cutting. The length of each groove was 10 mm, and the cutting toolpath was employed two times. The cut section was polished with 1000–2500 grit sandpaper and then soaked in anhydrous ethanol, followed by cleaning using an ultrasonic cleaner for effective observation. The cutting depth and the incision surface morphology were observed by an Ultra-Depth Three-Dimensional Microscope with DSX1000, as shown in Figure 3b. Each group of experiments was repeated three times, and the measured values were averaged three times.

## 3. Experimental Principles and Methods

### 3.1. Experimental Principle

Water-jet-guided laser processing technology is the use of fine water jets to guide the laser to the workpiece to achieve the precision processing of composite processing technology [21]. Its processing principle is shown in Figure 4. Since water and air have different refractive indices, when the laser beam after the focusing lens focuses on the liquid surface if the angle of incidence is less than the critical angle of reflection, the laser in the water jet will only occur in total reflection without refraction, which makes the laser beam completely restrained in the water jet transmission, coupled with the water jet as one. The water jet acts as a laser fiber and becomes the carrier of laser energy, wherein the laser is guided to the surface of the workpiece to achieve processing.

### 3.2. Experimental Method

In the water-jet-guided laser cutting Inconel 718 experiments, the cutting effect was influenced by several factors. According to the previous single-factor experimental results, we concluded that the variation of processing distance within a certain range has a negligible effect on the cutting depth. Unlike conventional laser energy beam focusing processing, the effective processing distance of water-jet-guided laser processing is the distance that the water jet can carry laser energy for stable transmission. After experimental verification, when the processing distance is too long (more than 200 mm), most of the laser energy is absorbed by the water jet in the transmission process, and the amount of absorption is related to the wavelength of the laser [22,23]. In addition, the serious scattering at the bottom of the water jet weakens the coupling effect of water and laser, and the energy density in the water jet fiber transmitted to the Inconel 718 surface cannot reach the melting threshold of the material, and in this case does not have the cutting ability. When the processing distance is controlled within 20 mm, the water jet within this range is stable and the transmitted laser energy can reach the melting threshold of Inconel 718 material. In this experiment, the processing distance was fixed to 15 mm.

On the basis of the previous experimental experience, the selected factors and levels were laser power 200~350 W, pulse frequency 4000~7000 Hz, feed speed 0.1~0.3 mm/s, and water pressure 1.2~1.8 MPa. The trend of cutting depth with four critical process parameters and its reasons were studied by orthogonal experimental result data, and the cutting quality, such as kerf neatness and kerf edge burn, were extensively studied. A mathematical prediction model about the process parameters on the cutting depth was established by the statistical RSM experimental data, and the validity of the model was verified by several experiments. The improved Taguchi orthogonal experimental design is shown in Table 2. The variables were normalized in the RSM experimental design and represented by coding form, and the factors and levels are shown in Table 3.

## 4. Experimental Results and Discussion

### 4.1. Results of Orthogonal Experiments

The Taguchi method relies on the value of the signal-to-noise index (S/N) regarding the mean squared deviation (MSD). It has the advantage of being able to take into account both the variance and the mean of the data and can effectively reflect the relationship between the influence of the factors and the results [24,25,26].
(1)The S/N equation is given by S/N=−10log(MSD)

The calculation of the mean squared deviation (MSD) depends on the characteristics of the index measured, and for the cutting depth measured in this experiment, we want it to be “as large as possible”.

The results of the orthogonal experiments on the depth of the cut is shown in Table 4. The ranking of the S/N values and mean extreme differences in the S/N response table shown in Table 5 and the mean main effects plot shown in Figure 5 describe the degree of influence of four factors on the cutting depth, in descending order: laser power, pulse frequency, water pressure, and feed speed.

The overall shape of the cross-section of the slit is shown in Figure 6, with a “V” shape. Because of the low thermal conductivity of Inconel 718, although the distribution of laser energy density in the water jet is uniform rather than Gaussian distribution, the hydrostatic pressure is in the middle part of the maximum pressure. Thus, the different degrees of laser energy impact and water cooling caused uneven heat dissipation between the middle part of the material and the surrounding area, and accompanied by the water jet on the scouring of the molten material, the overall shape of the slit presented a “V” shape.

#### 4.1.1. Influence of Feed Speed on Cut Quality

Figure 7 shows the average cutting depth at different feed speeds. When the feed speed increased from 0.1 mm/s to 0.2 mm/s and 0.3 mm/s, the average depth cutting decreased by 4.64 μm and 33.78 μm for multiple experiments. This was because the feed speed decided the length of time that the water jet carried the laser interaction with the material: the slower the feed speed, the longer the laser energy acts on the material, and the deeper the material reaches the melting threshold, the greater the cutting depth.

Figure 8 shows the surface morphology of the cut at different feed speeds. When the feed speed was 0.1 mm/s, the kerf shape was neat, the burns range was small, and the slag was less. When the feed speed increased to 0.2 mm/s and 0.3 mm/s, the burns of the kerf edge were gradually severe, the spatter and slag increased, and the overall machining quality decreased. The formation mechanism of the incision was the water jet carrying laser energy on the surface of Inconel 718 material, wherein the material rapidly heated up and heat transfer occurred towards the surroundings when the material temperature reached the melting threshold after melting, vaporization, and the formation of plasma. In between the laser energy pulse accompanied by the cooling and cleaning of the water jet, the heat of the ablation area was rapidly released, and most of the molten materials were washed clean by the water jet, and thus the incision was formed, and solidification occurred in part, forming slag and a recast layer on the surface of the incision. This shows that the feed speed also affected the degree of kerf ablation and the degree of slag residue. When the feed speed was larger, the water jet had a short cooling effect on the material, and the material surface was severely ablated. When the feed speed was reduced, the water jet acted on the material surface for longer. Moreover, more heat was able to be taken away during the laser pulse, thus reducing the heat transfer and heat accumulation in the material.

In summary, a feed speed of 0.1 mm/s can bring about a deeper cutting depth and high-quality surface topography.

#### 4.1.2. The Effect of Water Pressure on the Cutting

As can be seen from Figure 9, as the water pressure increased, the extent of ablation at the edge of the cut decreased, and the cut was neater. The reason was that as the water pressure increased, more heat was absorbed by the water jet in the material between laser energy pulses, which reduced the burns caused by heat accumulation on the material surface. At the same time, the increase in water pressure will bring more impact to the material and remove more slag from the edge of the cut. Therefore, the water pressure had an important effect on the formation of the material surface morphology.

The average cutting depth at different water pressures is shown in Figure 10. It can be seen that when the water pressures were 1.2 MPa, 1.4 MPa, and 1.6 MPa, the average depths of cuts were gradually increased to 556.6 μm, 584.8 μm, and 597.5 μm, respectively. The reason was that the high-speed water jet can enhance the removal of molten material from the groove. Moreover, the stable water jet was also able to achieve higher quality water-laser coupling, which can transmit more laser energy to the depth of the cut. When the water pressure was 1.8 MPa, the cutting depth decreased. The excessive water jet velocity during the laser energy pulse led to the absorption of most of the energy transferred by the laser to the material by the water jet, and the high-pressure water jet at the depth of the cutting seam was severely scattered. Thus, more laser energy cannot be transferred to the depth of the material cutting seam, resulting in a decrease in the heat accumulation of the material at the depth of the cutting seam, and the material part reached or was unable to reach the melting threshold. Therefore, the cutting depth showed a downward trend when the water pressure was too high.

#### 4.1.3. Effect of Laser Power and Pulse Frequency on the Cutting

Figure 11 shows the cross-section of the cutting depth for different laser powers and pulse repetition frequencies (excluding 1.8 MPa water pressure). We can see from the data in Figure 12 that the cutting depth gradually increased with increasing laser power and decreased with increasing laser pulse frequency. The reason was that the amount of laser power and pulse frequency determined the amount of laser energy density—the higher the laser energy density, the stronger the ability to remove the material.

When the laser power is fixed, the single pulse energy can be calculated by the following formula:(2)$$\mathrm{Ep}=\frac{\mathrm{laser}\;\mathrm{pwer}}{\mathrm{pulse}\;\mathrm{frequance}}$$

Ep: energy of a single pulse (J).

When the laser power is 280 W and the pulse frequencies are 4000 Hz, 5000 Hz, 6000 Hz, and 7000 Hz, the single pulse energies transmitted to the material, in theory, are 0.070 J, 0.056 J, 0.047 J, and 0.040 J, respectively. Considering the energy absorption of the water jet and the heat loss of energy on the material surface, the energy transmitted to the material is much lower than this value. As the pulse frequency increases, the single pulse energy density becomes smaller, while the water jet causes cooling to the material during the laser pulse interval. Therefore, a higher laser pulse frequency makes the material absorb less laser energy per unit time, causing shallower cutting depth. This is the same as the research results of Z et al. [27].

### 4.2. Response Surface Methodology Experimental Results

Response surface methodology is a mathematical method for fitting the relationship between the unknown variables and the objective function by using the multivariate quadratic regression equation to reasonably utilize the statistical results of the experimental data. It can simulate the experimental data as a surface showing the true situation and reflect the relationship between the unknown variables and the response values through the surface [28,29].

In the RSM experiments, a total of 29 sets of experiments were conducted according to the central combination experimental design. Five of the central experimental groups were used to estimate experimental error, and the remaining groups were used to analyze the relationship between factors and response value. Table 6 shows the RSM experiments and results regarding the cutting depth.

The data from the experimental results were fitted using Design-Expect to obtain the regression equation on the cutting depth:(3)$$\begin{array}{cc} \mathrm{Cutting}\;\mathrm{depth}\;= & 557.30-46.16\times A+122.01\times B-93.21\times C-56.98\times D\\ & +16.80\times \mathrm{AB}+14.60\times \mathrm{AC}-13.48\times \mathrm{AD}+3.90\times \mathrm{BD}\\ & +9.95\times \mathrm{CD}-99.75\times A^{2}+3.02\times B^{2}-7.35\times C^{2}\\ & -3.93\times D^{2} \end{array}$$

Figure 13 shows the residual normal distribution of the cutting depth prediction model. The residual value of the model was uniform in a straight line, indicating that the predicted value of the model for cutting depth was in good agreement with the actual value. Figure 14 shows the residual distribution between the actual value and the predicted value. It shows that the data were scattered and irregularly distributed above and below the average line, which means the developed model was recognized to be statistically significant.

#### 4.2.1. Results of the ANOVA for the Model

Table 7 shows the results of the ANOVA on the cutting depth prediction model. The table shows that the F value of the model was 26.28 and the value of “Prob > F” (*p*-value) was <0.0001, which indicates that the prediction model of the regression equation established between the cutting depth of and the four independent variables was highly significant and that the prediction model can effectively respond and predict the variation relationship between the cutting depth and the independent variables. The *p*-value of model “lack of fit” was 0.1524 > 0.05, which indicates that the lack of fit was not significant. The multivariate correlation coefficient “R-squared” value was 0.9633, and the adjusted multivariate correlation coefficient “Adj R-Squared” value was 0.9267, indicating that the prediction model explained 92.67% of the response values and had a high accuracy of prediction.

Similar to the analysis of the previous orthogonal experiment results, the RSM showed that the laser power had the most impact on the cutting depth among the four factors, and the F value was 156.34, followed by the pulse frequency, and the F value was 91.24. It is proved once again that the laser power and pulse frequency are crucial to the cutting depth in the water-jet-guided laser cutting Inconel 718 experiment.

#### 4.2.2. Response Surface Analysis of Influencing Factors

Figure 15 shows the 3D surface plot (Figure 15a) and contour plot (Figure 15b) of the effect of the interaction term BC on the response value, i.e., the pattern of laser power and pulse frequency on the cutting depth. It can be seen that when the laser power was constant at the center point 275 W, with the increase in laser pulse frequency (4000 Hz, 5500 Hz, 7000 Hz), the cutting depth gradually decreased. When the laser power was 200 W and the pulse frequency was 7000 Hz, the single-pulse energy is the smallest, the cutting ability of the material was the weakest, and the cutting depth was the shallowest. Therefore, the way to obtain a deeper cutting depth was to select the laser power of 350 W and pulse frequency of 4000 Hz.

Figure 16 shows the response surface plot (Figure 16a) and contour plot (Figure 16b) of the effect of the interaction term AB on the response value, i.e., the pattern of water pressure and laser power on the cutting depth. When the laser power was fixed at the center point of 275 W, the cutting depth increased first and then decreased with the increase in water pressure (1.2 MPa, 1.5 MPa, 1.8 MPa). This also had an important relationship with the stability of the water jet. When the water pressure is small, partial molten materials cannot be excluded at the depth of the slit, and there is no way to remove deeper materials. When the water pressure is too large but unstable, the laminar flow state of the water jet is destroyed, which leads to a significant weakening of the coupling effect and a serious loss of laser energy in the transmission process, and thus a deeper cutting depth cannot be achieved [30]. It can be seen from the response surface Figure 16a that when the water jet pressure was above 1.65 MPa, the predicted cutting depth reached the optimal.

Figure 17a,b shows the influence of water pressure and feed speed on cutting depth. When the water pressure was fixed at 1.5 MPa, the cutting depth increased with the decrease in feed speed. When the feed speed was 0.1 mm/s, the maximum predicted cutting depth was achieved. The cutting speed of 0.1 mm/s made the laser energy act on the Inconel 718 material for the longest time, which increased the removal depth of the Inconel 718 material.

#### 4.2.3. Optimal Value Prediction and Experimental Validation

The experimental results in Table 6 were simulated by Design-Expect for data to optimize the response values, and the optimal processing parameters were obtained: laser power 350 W, pulse frequency 4000 Hz, water pressure 1.66 MPa, and feed speed 0.1 mm/s, at which time the predicted cutting depth was 795.6 μm.

In order to verify the prediction accuracy of the prediction model, three cutting experiments were carried out with the optimal parameters. The results of predicted values and experimental results are shown in Figure 18. The cutting depths were 722.7 μm, 759.2 μm, 774.4 μm, and the average depth was 752.1 μm. As shown in Table 8, the error between the actual value measured in the experiment and the predicted value obtained by the prediction model was 5.5%. This shows that the model has a high predictive ability for cutting depth.

Figure 19 shows the comparison of Inconel 718 cut by WJGL (Figure 19a) and LBM (Figure 19b). The incision burns under WJGL were greatly reduced. Figure 20b is the use of the same optimal process parameters, increasing the number of cutting toolpath (from two to four times) cutting results, wherein the thickness of 1 mm of the Inconel 718 plate was cut through.

## 5. Conclusions

In this study, the influence of critical parameters on the cutting Inconel 718 effect of WJGL was studied, and a mathematical model for optimizing the cutting depth was established. According to the experimental results obtained by WJGL cutting Inconel 718, the following conclusions were drawn:In the experiment of the water-jet-guided laser cutting Inconel 718, water played an important role. It was not only able to conduct laser energy, but also cooled the cutting and took away the slag. Compared with traditional cutting, this technology can bring about a higher cutting quality.The influence of critical parameters on cutting quality was studied by an orthogonal experiment. The experimental results show that the laser power had the greatest influence on the cutting quality, followed by the pulse frequency. The laser power and pulse frequency together determined the energy of a single pulse.The regression models of water pressure, laser power, pulse frequency, and feed rate on cutting depth were established by the response surface method. The results show that the model can predict 92.67% response value. The influence of different parameters on cutting depth was analyzed, and the order of factors affecting cutting depth was laser power > pulse frequency > feed speed > water pressure. Finally, the cutting experiment was carried out to verify the process parameters obtained by the maximum cutting depth. The maximum cutting depth was 774.4 μm, and the error with the predicted value was 5.5%, which proved the validity of the model.The water-jet-guided laser processing technology has great development potential. We will continue to explore the impact of high-pressure water jets on processing efficiency in our next research.

## Figures and Tables

**Figure 1 micromachines-14-00234-f001:**
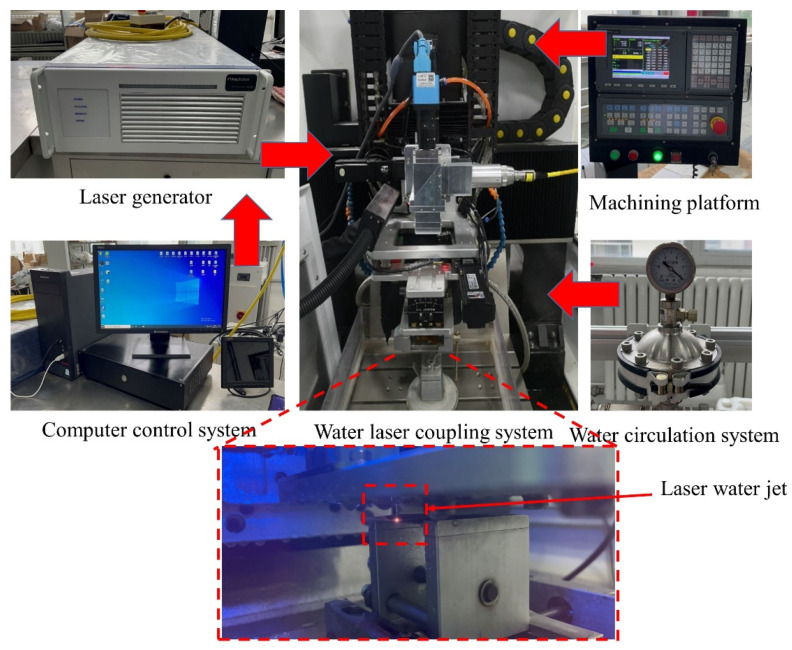
Water-jet-guided laser processing equipment.

**Figure 2 micromachines-14-00234-f002:**
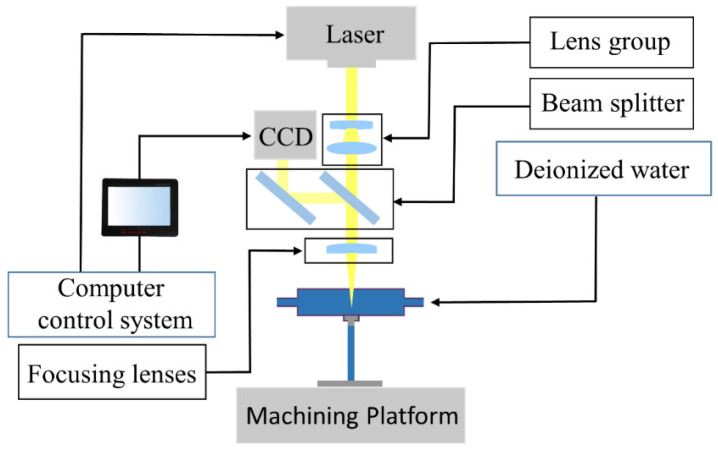
Schematic diagram of the water-jet-guided laser processing platform.

**Figure 3 micromachines-14-00234-f003:**
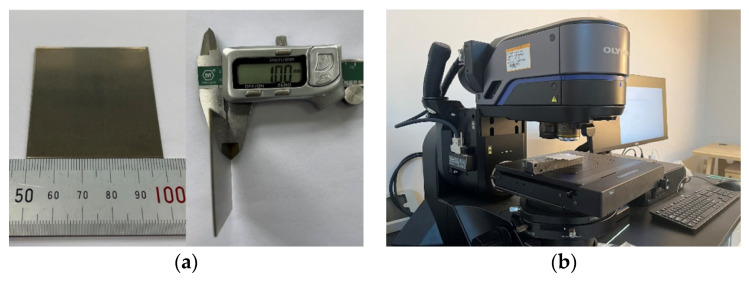
Sample and measurement setup. (**a**) Inconel 718 sample plate. (**b**) Ultra-Depth Three-Dimensional Microscope.

**Figure 4 micromachines-14-00234-f004:**
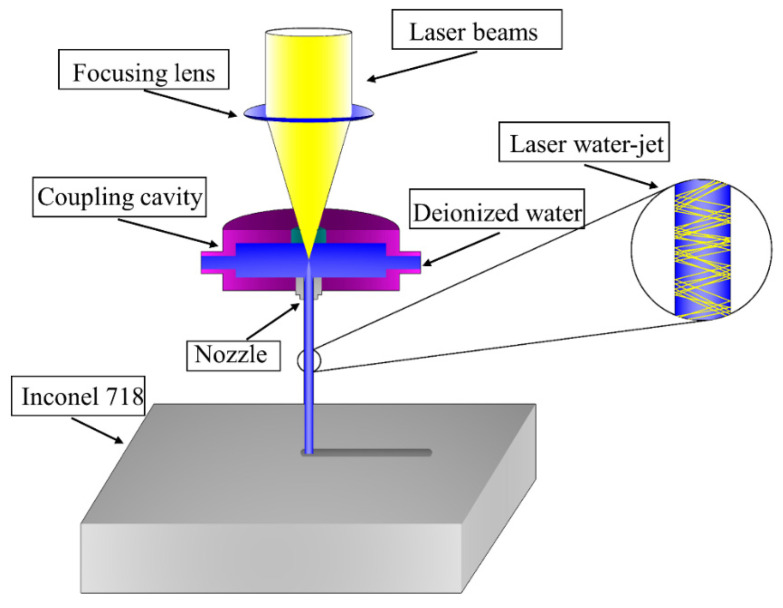
Principle of the water-jet-guided laser processing.

**Figure 5 micromachines-14-00234-f005:**
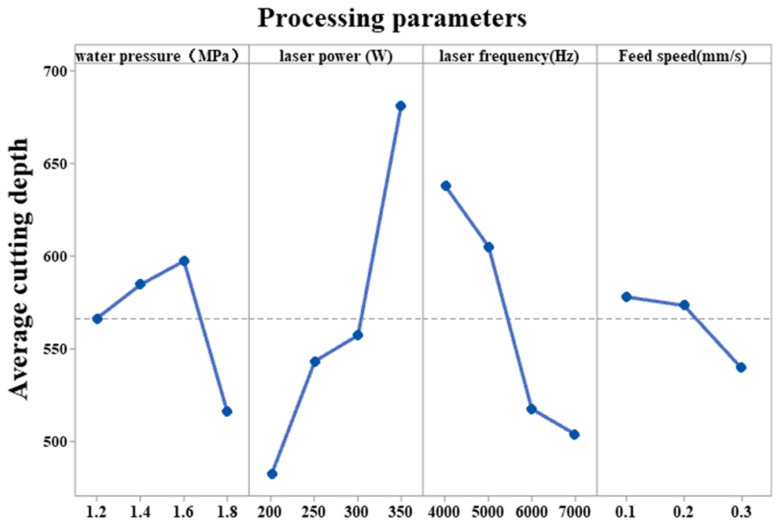
Main effect diagram of the mean value of cutting depth.

**Figure 6 micromachines-14-00234-f006:**
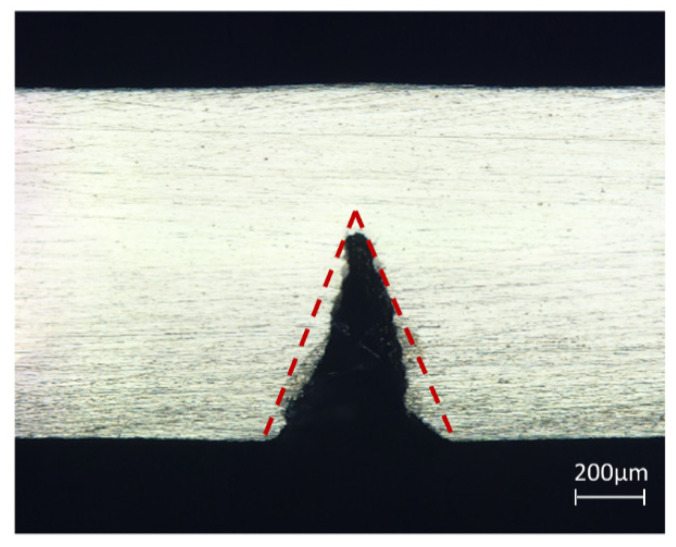
Cut section morphology.

**Figure 7 micromachines-14-00234-f007:**
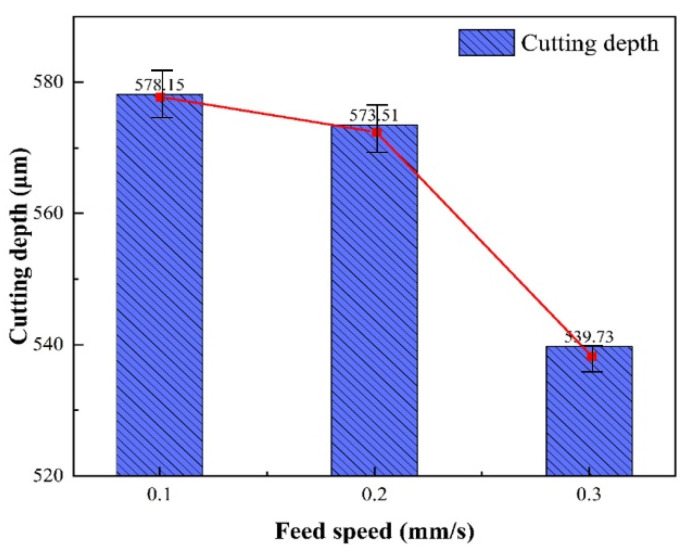
Average cutting depth at different feed speeds.

**Figure 8 micromachines-14-00234-f008:**
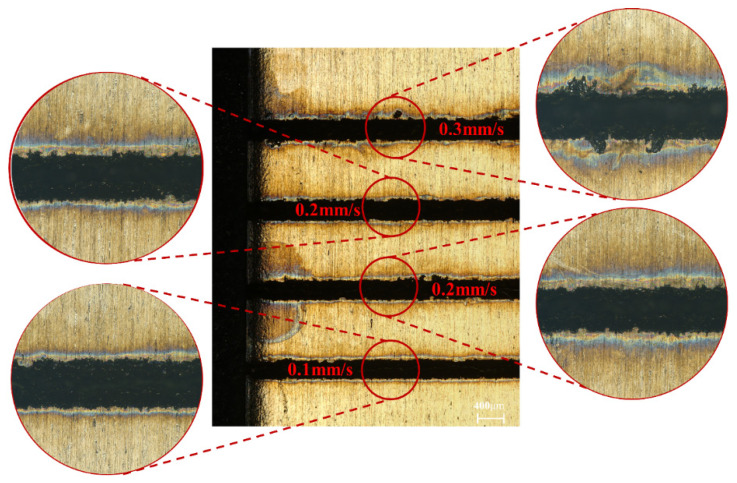
Cut surface morphology at different feed speeds.

**Figure 9 micromachines-14-00234-f009:**
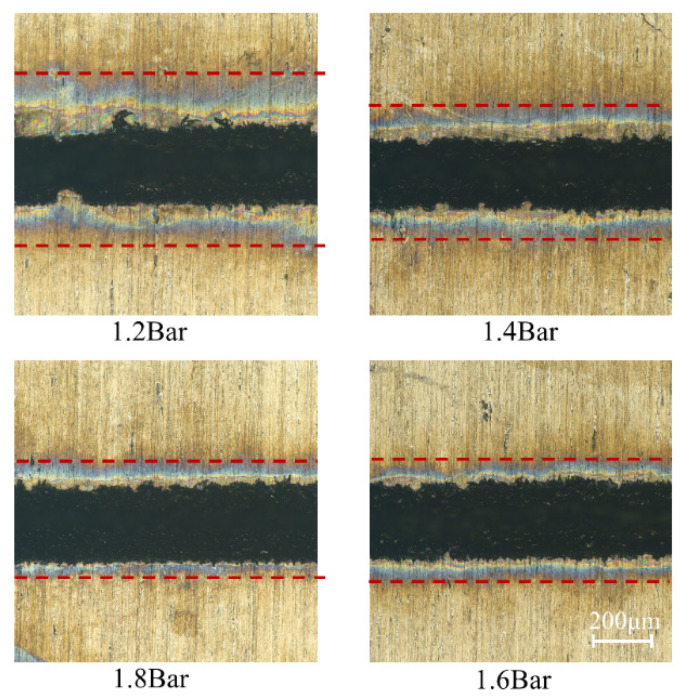
Cutting surface morphology under different water pressures.

**Figure 10 micromachines-14-00234-f010:**
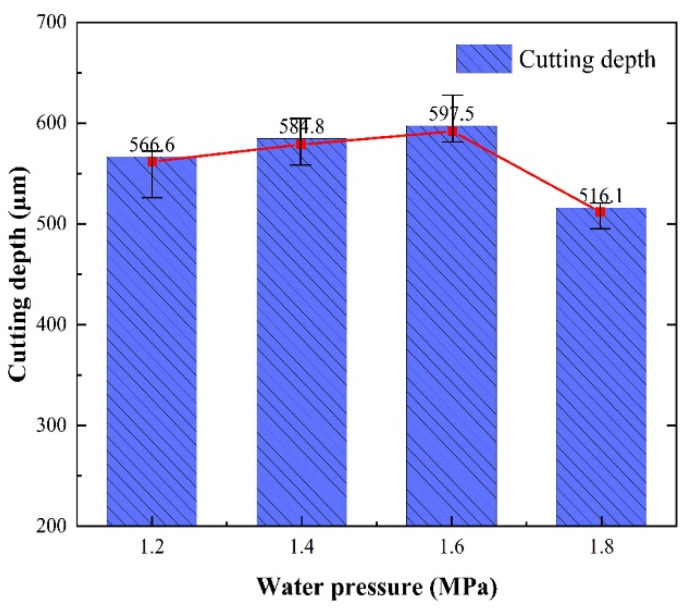
Average cutting depth at the different water pressures.

**Figure 11 micromachines-14-00234-f011:**
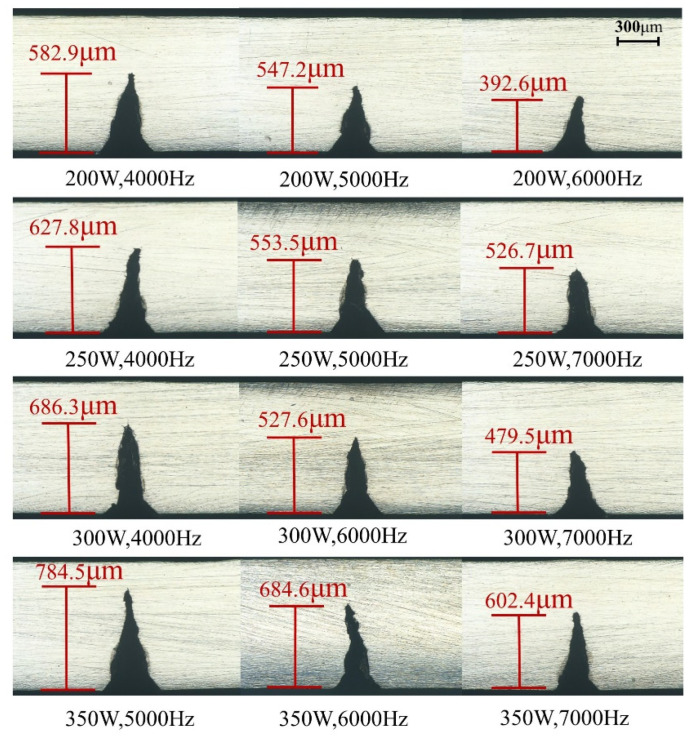
Cutting depth at different laser powers and pulse frequencies (excluding 1.8 MPa water pressure).

**Figure 12 micromachines-14-00234-f012:**
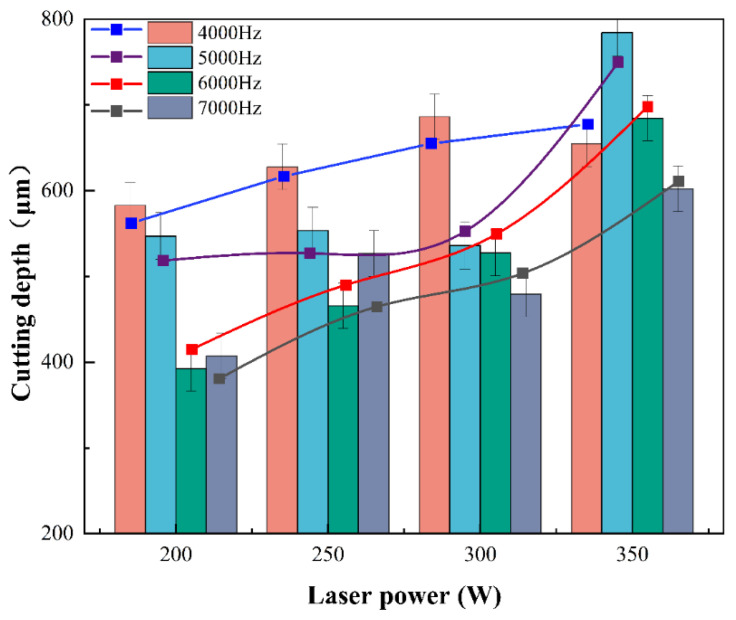
Average cutting depth at different laser powers and pulse frequencies.

**Figure 13 micromachines-14-00234-f013:**
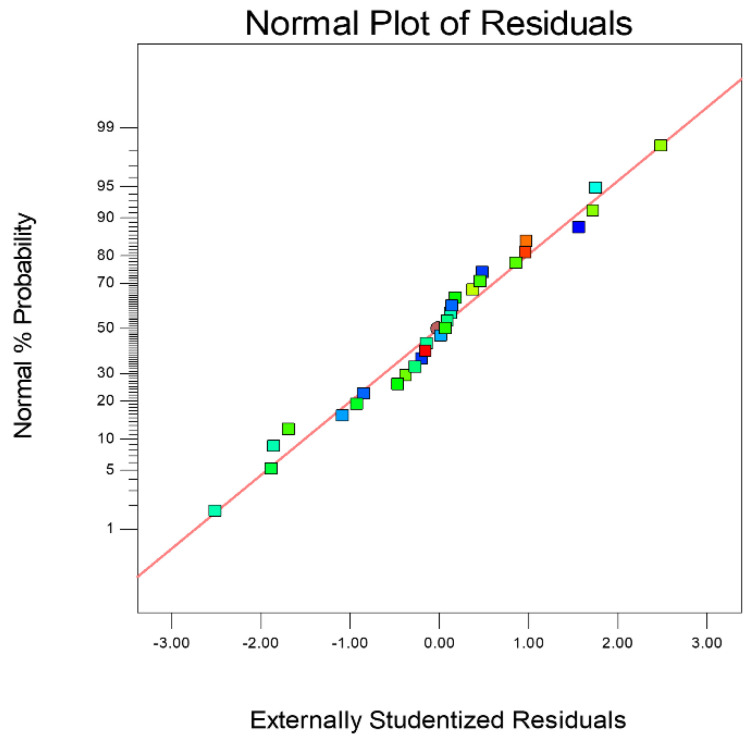
Normal plot of residuals.

**Figure 14 micromachines-14-00234-f014:**
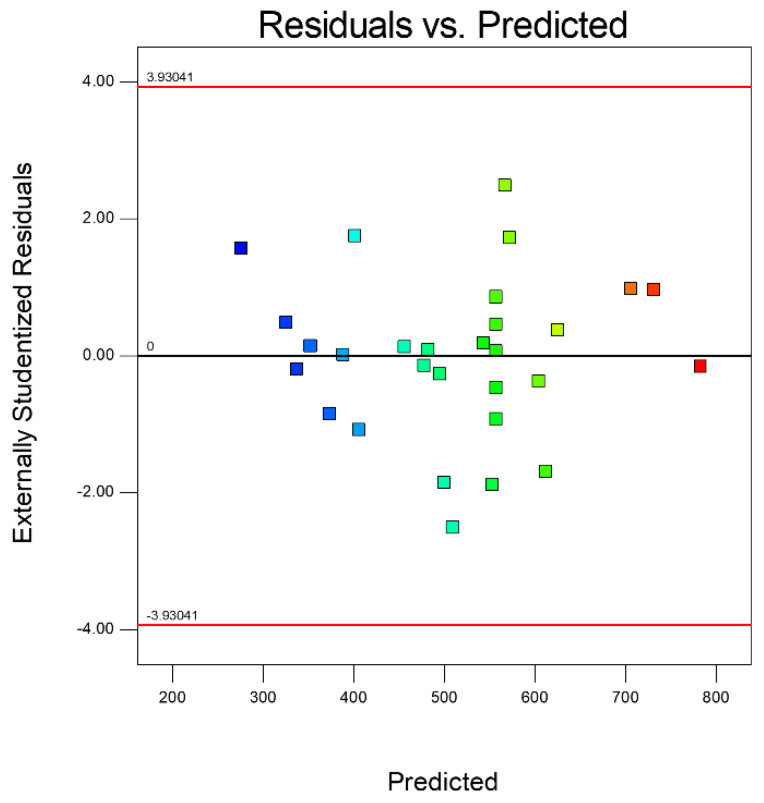
Residual distribution of actual value and predicted value.

**Figure 15 micromachines-14-00234-f015:**
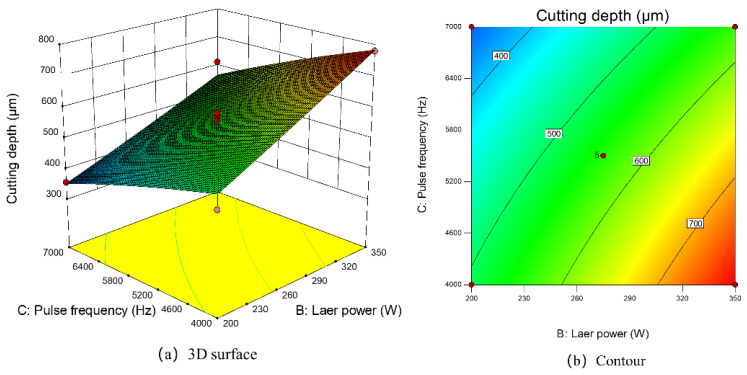
The 3D surface and contour map of the influence of laser power and pulse frequency on cutting depth.

**Figure 16 micromachines-14-00234-f016:**
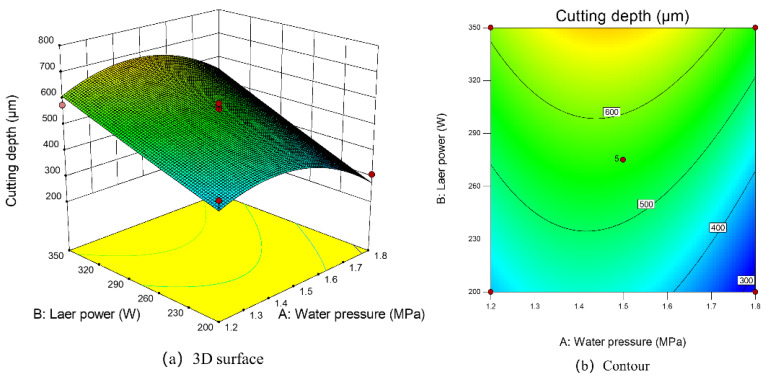
The 3D surface and contour map of the influence of laser power and water pressure on cutting depth.

**Figure 17 micromachines-14-00234-f017:**
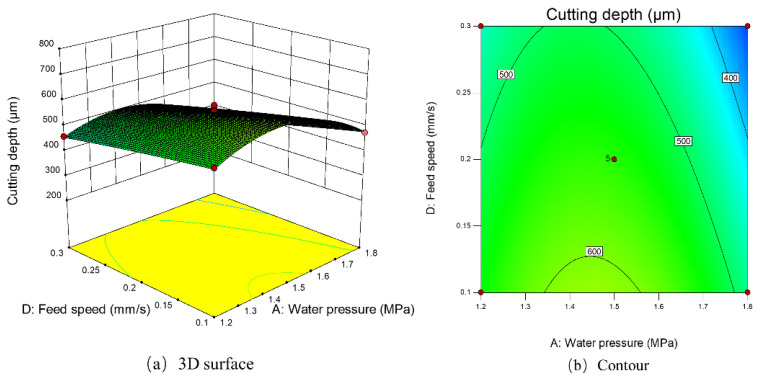
The 3D surface and contour map of the influence of feed speed and water pressure on cutting depth.

**Figure 18 micromachines-14-00234-f018:**
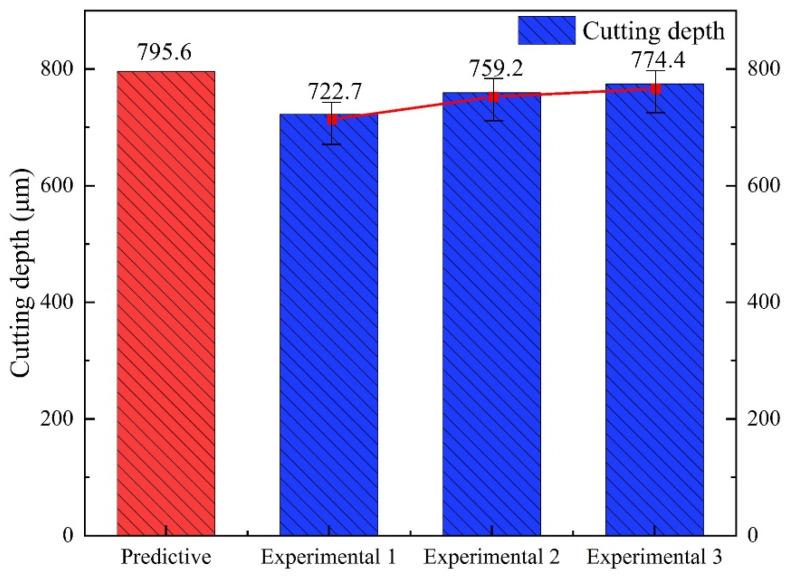
The results of predicted values and experimental results.

**Figure 19 micromachines-14-00234-f019:**
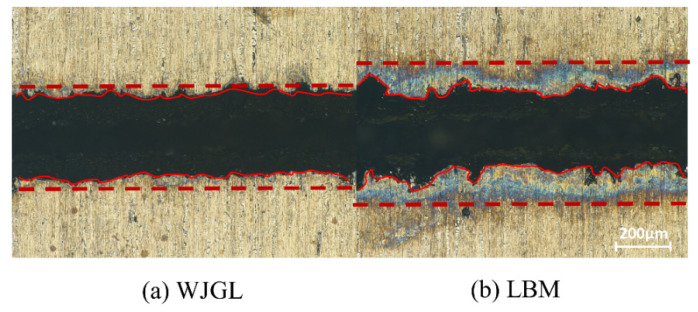
Comparison between WJGL and LBM.

**Figure 20 micromachines-14-00234-f020:**
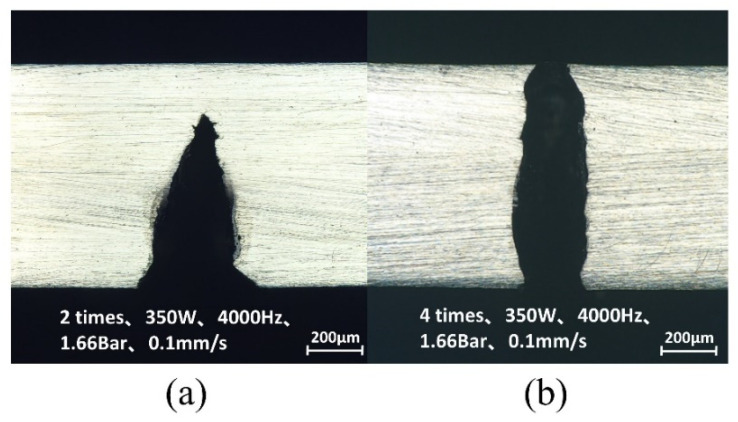
Cutting effect with different cutting toolpath times after optimization.

**Table 1 micromachines-14-00234-t001:** Chemical composition of Inconel 718 (wt %) [3].

Element	Ni	Cr	Fe	Nb	Mo	Ti	Al	Co	C	Mn	S	P	Si	B	Cu
Composition	54.2	18.4	17.3	5.2	2.9	0.98	0.5	0.3	0.02	0.08	<0.01	0.012	0.06	0.002	0.08

**Table 2 micromachines-14-00234-t002:** Taguchi orthogonal experiment factor and level table.

Factors	Unit	Level			
		1	2	3	4
Water jet pressure	MPa	1.2	1.4	1.6	1.8
Laser power	W	200	250	300	350
Laser pulse width	mm	4000	5000	6000	7000
Feed speed	mm/s	0.1	0.2	0.3	-

**Table 3 micromachines-14-00234-t003:** Response surface methodology experimental factor level and coding value corresponding table.

Factors	Unit	Coding Level		
		−1	0	+1
Water jet pressure	MPa	1.2	1.5	1.8
Laser power	W	200	275	350
Pulse frequency	Hz	4000	5500	7000
Feed speed	mm/s	0.1	0.2	0.3

**Table 4 micromachines-14-00234-t004:** Taguchi orthogonal experimental dataset.

Trial	Water Jet Pressure (MPa)	Laser Power (W)	Pulse Frequency (Hz)	Feed Speed (mm/s)	Cutting Depth (μm)
1	1.2	200	4000	0.1	582.9
2	1.2	250	5000	0.2	553.5
3	1.2	250	6000	0.2	527.6
4	1.2	350	7000	0.3	602.4
5	1.4	200	5000	0.2	547.2
6	1.4	250	4000	0.3	627.8
7	1.4	300	7000	0.1	479.5
8	1.4	350	6000	0.2	684.6
9	1.6	200	6000	0.3	392.6
10	1.6	250	7000	0.2	526.7
11	1.6	300	4000	0.2	686.3
12	1.6	350	5000	0.1	784.5
13	1.8	200	7000	0.2	407.4
14	1.8	250	6000	0.1	465.7
15	1.8	300	5000	0.3	536.1
16	1.8	350	4000	0.2	654.8

**Table 5 micromachines-14-00234-t005:** S/N response table.

Level	Water Jet Pressure (MPa)	Laser Power (W)	Pulse Frequency (Hz)	Feed Speed (mm/s)
1	55.05	53.54	56.08	55.05
2	55.26	54.65	55.53	55.06
3	55.23	54.84	54.10	54.50
4	54.12	56.63	53.96	-
Delta	1.14	3.09	2.12	0.55
Order	3	1	2	4

**Table 6 micromachines-14-00234-t006:** RSM experimental results.

Trial	Water Jet Pressure (MPa)	Laser Power (W)	Pulse Frequency (Hz)	Feed Speed (mm/s)	Cutting Depth (μm)
1	1.2	200	5500	0.2	437.2
2	1.5	275	4000	0.3	607.5
3	1.8	275	5500	0.1	474.7
4	1.5	350	5500	0.3	633.8
5	1.5	350	5500	0.1	752.6
6	1.8	275	5500	0.3	332.6
7	1.5	200	5500	0.3	354.8
8	1.2	275	7000	0.2	388.9
9	1.5	275	5500	0.2	559.4
10	1.2	350	5500	0.2	577.3
11	1.5	350	4000	0.2	779.4
12	1.5	200	5500	0.1	489.2
13	1.5	200	7000	0.2	355.7
14	1.5	350	7000	0.2	613.4
15	1.8	275	7000	0.2	336.3
16	1.5	275	5500	0.2	571.6
17	1.8	350	5500	0.2	515.4
18	1.5	275	4000	0.1	727.5
19	1.8	275	4000	0.2	484.7
20	1.8	200	5500	0.2	308.1
21	1.5	200	4000	0.2	462.8
22	1.5	275	7000	0.3	382.3
23	1.5	275	5500	0.2	529.2
24	1.5	275	7000	0.1	462.5
25	1.2	275	5500	0.3	459.2
26	1.5	275	5500	0.2	542.8
27	1.2	275	4000	0.2	595.7
28	1.5	275	5500	0.2	583.5
29	1.2	275	5500	0.1	547.4

**Table 7 micromachines-14-00234-t007:** Variance analysis of cutting depth regression model.

Source	Sum of Squares	df	Mean Square	F	Prob > F	
Model	4.204 × 10^5^	14	3.003 × 10^4^	26.28	<0.0001	Significant
A-water pressure	2.557 × 10^4^	1	2.557 × 10^4^	22.38	0.0003	
B-laser power	1.786 × 10^5^	1	1.786 × 10^5^	156.34	<0.0001	
C-pulse frequency	1.043 × 10^5^	1	1.043 × 10^5^	91.24	<0.0001	
D-feed speed	3.895 × 10^4^	1	3.895 × 10^4^	34.09	<0.0001	
AB	1.129 × 10^3^	1	1.129 × 10^3^	0.99	0.3371	
AC	8.526 × 10^2^	1	8.526 × 10^2^	0.75	0.4022	
AD	7.263 × 10^2^	1	7.263 × 10^2^	0.64	0.4386	
BC	8.673 × 10^2^	1	8.673 × 10^2^	0.76	0.3983	
BD	0.608 × 10^2^	1	0.608 × 10^2^	0.053	0.8208	
CD	3.960 × 10^2^	1	3.960 × 10^2^	0.35	0.5654	
A^2^	6.455 × 10^4^	1	6.455 × 10^4^	56.49	<0.0001	
B^2^	0.592 × 10^2^	1	0.592 × 10^2^	0.052	0.8232	
C^2^	3.508 × 10^2^	1	3.508 × 10^2^	0.31	0.5882	
D^2^	1.001 × 10^2^	1	1.001 × 10^2^	0.088	0.7715	
Residual	1.600 × 10^4^	14	1.143 × 10^3^			
Lack of fit	1.410 × 10^4^	10	1.410 × 10^3^	2.98	0.1524	Not significant
Pure error	1.895 × 10^3^	4	4.738 × 10^2^			
Cor total	4.364 × 10^5^	28				
R-squared = 0.9633				Adj R-squared = 0.9267		

**Table 8 micromachines-14-00234-t008:** Experimental results of the optimal parameters.

Experiment 1	Experiment 2	Experiment 3	Mean Experiment	Predictive	Error
722.7 μm	759.2 μm	774.4 μm	752.1 μm	795.6 μm	5.5%

## Data Availability

The datasets used or analyzed during the current study are available from the corresponding author upon reasonable request.

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
