# Peer review of "Modeling and Prediction of Water-Jet-Guided Laser Cutting Depth for Inconel 718 Material Using Response Surface Methodology"

_micromachines, 2023, doi:10.3390/mi14020234_

Round 1

Reviewer 1 Report

This paper emphasizes water plays an important role in Water jet Guided Laser Cutting. However, the water pressure in this paper is very small, and there is no picture display of water beam stability experiment and water jet guided laser processing state. From my understanding, the water beam did not play the role of guiding the laser, but only achieved the water cooling effect. Therefore, the research content does not seem to be consistent with the topic. That the laser power has the great influence on the cutting quality, followed by the pulse frequency It shows that it is mainly the role of laser, and the effect of water conduction is not reflected.

Reviewer 2 Report

1. Abstract should be given as more interesting. Express at least one of the main aspects and features of the paper.

2.  At the end of the Introduction section, it would be better to add the paper's organization in different sections.

3.  Response surface methodology has been adopted in several research works also. Authors are invited to cite recent published works.

4.  Include the photograph of the cutting depth measurement setup.

5.  Provide the suitable citation of literature in Table 1 for chemical composition of Inconel 718.

6. Given that the manuscript is heavily based on experimental work and measurements, it is vital for the authors to report on the method(s) to improve measurement reliability. The methods/measures that have been taken to minimize experimental errors and improve reliability should be included in the experimental setup.

7.The discussion related to various diagnostic tests are essential for the validation of proposed RSM model,

8. Provide the error bar in Figure 7 &10.

9.Further, results and analysis of experiments should be compared with previous researchers by citing references.

10. In the statistical analysis (Table 7), mention the standard F-value obtained from the F-distribution table while comparing with the computed value.

11. Figure 11: Scale bar for optical image is missing.

12. Provide the information of RSM optimization results by RAMP function plot and Bar plot.

13. Improve the conclusion with scope for future work.

14.Manuscript must be presented in highlight the contribution, and applicability of the work.

15.  Please check the manuscript for wrong choice of words, grammatical errors and incoherent sentence structure.

Reviewer 3 Report

Please check the space between the quantity value and the unit. (lines 93 to 103)

Round 2

Reviewer 1 Report

The innovation of this paper is not obvious. It is suggested to change the investment.

Reviewer 2 Report

Authors have made significant changes in the revised manuscript. Hence, consider the manuscript for publication in its present form.